# Development and validation of a health education module for parents of schoolchildren with overweight and obesity in the UAE

Heba S. M. Mustafaalsaafin[1,2], Hamid Jan B. Jan Mohamed[2], Hafzan Yusoff[2], Mona Hashim[1,3], Hadia Radwan[1,3], Leila Cheikh Ismail[1,3,4], Maysm N. Mohamad[5], Heba F. Almassri[2], Hayder Hasan[1,3]*

1 Research Institute of Medical and Health Sciences, University of Sharjah, Sharjah, United Arab Emirates, 2 Nutrition Program, School of Health Sciences, Universiti Sains Malaysia, Kelantan, Malaysia, 3 Department of Clinical Nutrition and Dietetics, College of Health Sciences, University of Sharjah, Sharjah, United Arab Emirates, 4 Nuffield Department of Women's and Reproductive Health, University of Oxford, Oxford, United Kingdom, 5 Department of Nutrition and Health, College of Medicine and Health Sciences, United Arab Emirates University, Al Ain, United Arab Emirates

* haidarah@sharjah.ac.ae

## Abstract

### Background

Childhood overweight and obesity continue to rise globally and in the United Arab Emirates (UAE), underscoring the need for accessible, evidence-based parental education tools.

### Objective

This study aims to develop and validate a health educational module (HEM) tailored for parents of schoolchildren with overweight and obesity in the UAE.

### Methods

The study was conducted in two phases: module development, and content and face validation. The module development involved an extensive review of national and international guidelines and previous research, followed by the design of infographic-based educational messages. Content and Face Validity were performed by six nutrition experts and 16 parents, respectively, using the Content Validity Index (CVI) and Face Validity Index (FVI).

### Results

The final HEM consisted of 25 infographic messages covering dietary intake, physical activity, behavior change, and family engagement. The module achieved excellent content validity (S-CVI/Ave = 0.97) and face validity (S-FVI/Ave = 0.99). Participant

**Data availability statement:** All relevant data are within the manuscript and its Supporting information files.

**Funding:** HH received funding from the Office of the Vice-Chancellor Research and Graduate Studies at the University of Sharjah, grant number (2201050780) VCRG/R. 447/2022. https://www.sharjah.ac.ae/Discover-UoS/VCRGS The funders had no role in study design, data collection and analysis, decision to publish, or preparation of the manuscript.

**Competing interests:** The authors have declared that no competing interests exist.

feedback resulted in language refinement, improved illustrations, and removal of outdated elements such as the Food Pyramid.

## Conclusion

The validated HEM is culturally relevant, parent-friendly, and scientifically grounded. It offers a structured, high-quality educational tool suitable for use in research, school-based initiatives, and public-health programs aimed at reducing childhood obesity in the UAE.

## Introduction

Childhood obesity is a global public health concern with far-reaching implications for the health and well-being of affected children and society as a whole. It has become a pressing global health concern, as its prevalence has reached alarming proportions in recent decades [1]. The worldwide prevalence of children and adolescents aged 5–19 affected by overweight and obesity has seen a substantial increase, soaring from a mere 8% in 1990 to 20% in 2022 [1]. The UAE, like many other countries, is grappling with the rising prevalence of schoolchildren with overweight and obesity that is becoming a multifaceted issue requiring comprehensive interventions [2–4]. The 2019 national survey on obesity among school-age children (5–17 years old) in the UAE revealed that 17.35% of this demographic were living with obesity [4].

Childhood obesity is associated with multiple physical, psychological, and social consequences, ranging from cardiovascular diseases and diabetes mellitus to low self-esteem and stigmatization [5]. The psychosocial repercussions of obesity during childhood and adolescence are significant, impacting school performance and overall quality of life. This is exacerbated by the presence of stigma, discrimination, and bullying [6]. Moreover, children living with obesity face an increased likelihood of carrying this condition into adulthood, along with an increased risk of developing noncommunicable diseases later in life [7]. The ramifications of this public health issue extend far beyond individual well-being, as they place a substantial burden on healthcare systems and society as a whole [8].

Addressing childhood obesity necessitates a holistic approach that encompasses not only the children themselves but also their families and communities [9]. Parents, as primary caregivers and role models, play a pivotal role in shaping children's dietary habits and physical activity patterns [10,11]. Empowering parents with the knowledge and skills needed to promote healthier lifestyles within the family context has been recognized as a crucial strategy for combating childhood obesity [11,12].

Health education interventions tailored for parents of schoolchildren affected by overweight and obesity have emerged as a promising strategy for addressing childhood obesity [13]. Such interventions hold the potential to enhance parental awareness, improve decision-making regarding nutrition and physical activity, and ultimately contribute to the successful management of childhood obesity [13,14].

Nevertheless, these interventions must be contextually relevant and culturally sensitive to ensure their effectiveness within specific populations [15].

While many initiatives have shown success in diverse cultural settings, it is essential to recognize the unique sociocultural context of the UAE. Cultural differences, dietary preferences, and lifestyle practices vary significantly across regions, influencing the effectiveness of internationally designed interventions within the specific culture of the UAE [16]. Therefore, there is a need for an intervention that is crafted with a deep understanding of local traditions, dietary habits, and family dynamics that would resonate better with the population and increase the likelihood of successful adoption and maintenance of positive health outcomes.

Despite the growing number of childhood obesity interventions in the UAE, existing research indicates several limitations that restrict their effectiveness [2–4,17]. Many studies highlight parental influence but do not offer structured, evidence-based tools designed specifically for parents [10–12,17]. Moreover, previous interventions often lack cultural adaptation, which is essential for effectiveness in the UAE context, where dietary habits and family dynamics differ significantly from Western settings [15,16]. Existing programs also tend to overlook behavioral-change theories or provide educational materials that are either too technical for the general population or too fragmented for practical use [3,11,17]. Only a few studies in the region have attempted to develop parent-focused materials, and even fewer have included systematically validated, parent-friendly modules that integrate nutrition, physical activity, family engagement, and behavior-change strategies [11,17]. These gaps underscore the need for a comprehensive, rigorously validated, culturally appropriate module tailored specifically to parents, an area this study aims to address.

This research paper presents the development and validation of a HEM designed specifically for the parents of school-children with overweight and obesity in the UAE. Recognizing the unique sociocultural context of the UAE and building upon the existing body of research on parental interventions in childhood obesity, this study aims to fill a critical gap in the literature by providing an evidence-based, culturally adapted, and validated resource for parents seeking to address childhood obesity within their families. Through a rigorous process of development and validation, we endeavor to contribute to the growing body of knowledge on effective strategies for combatting childhood obesity in the UAE and, by extension, inform similar initiatives worldwide.

## Methods

### Study design and ethics

This study consisted of two main phases: 1. Health education module development, and 2. Health education module validation. It was conducted in accordance with ethical guidelines, and it adhered to the ethical principles outlined in the Declaration of Helsinki. Ethical approval was obtained from the Research Ethics Committee of the University of Sharjah, Sharjah, UAE (Reference Number: REC-22-09-05-01). The protocol of the study was also reviewed and approved by the Ministry of Education and Sharjah Private Education Authority in the UAE. Written informed consent was obtained from the participants before involving them in the study. The recruitment of the participants started on Dec. 06, 2022, and ended on Jan. 19, 2023. The participating subjects consisted of six nutrition-related experts for content validation and sixteen mothers of school-aged children for face validation.

**Health education module development phase.** The development of the health education module step began with a comprehensive background literature review and a collection of relevant resources to be used in the development of the module. Nine resources were referred to and focused on during this step including national guidelines [18,19], international guidelines [20–24], and literature [25,26].

The module was structured to be a tool for health educators to use and a systematic guideline for parents to learn how to guide a healthy lifestyle for their children. The module was written in Arabic language with the use of layman's terms to make it a reader friendly tool and to communicate the information effectively. However, an English version was created as well for the benefit of the non-Arab residents of the UAE and for others worldwide to use with modifications according to

their needs. The module was created to address key topics such as nutrition, physical activity, behavior change, and family engagement. It was structured in the form of infographic messages addressing parents, where each message delivers a specific idea. Culturally appropriate examples, and attractive and colourful illustrations were integrated into the module. The content of the module was based on the national and international guidelines as well as the other resources that were reviewed in the background reading.

The development of the HEM was informed by the Health Belief Model (HBM), which emphasizes perceived susceptibility, benefits, and the reduction of barriers to action. These theoretical constructs guided the framing of messages to increase parents' awareness of childhood obesity risks, highlight the benefits of preventive behaviors, and empower them to take practical steps within their cultural and social context. The content was grounded in national and international dietary and physical activity guidelines, ensuring both scientific accuracy and contextual relevance.

The module was created and formatted with multiple factors taken into consideration including: 1. Setting reasonable and achievable goals. 2. Focusing on the long-term behavioral changes. 3. Diversification of the methods parents can use with their children including observational learning and role modelling, decision making involvement of the children, positive reinforcements, household adjustments and whole family involvement.

**Health education module validation phase. Content and face validation by experts:** A panel of healthcare experts in the field of nutrition was invited to evaluate the module. The panel consisted of two dietitians, two nutrition educators and two nutrition research experts. It is recommended that there should be at least six members on the panel to avoid chance agreement [27].

Content validation was carried out using a content validation form. The validation form was adapted from an instrument developed by Castro et al. [28]. The form consisted of seven aspects for content and face evaluation with 41 items under all of them together. The aspects included: scientific accuracy, content, literary presentation, illustrations, sufficiently specific and understandable material, legibility and display characteristics, and quality of information. The experts reviewed each item and gave it a rate of 1–4 judging the level of relevancy of each statement to the module where 1 was for "not relevant"; 2 for "somewhat relevant"; 3 for "quite relevant"; and 4 for "highly relevant". The form also included four open-ended questions for the experts to answer regarding what they liked and disliked and what should be added or reviewed in the module.

The experts were also asked to review each message in the module and rate it on a scale of 1–4 to indicate how relevant each one is regarding the current guidelines and the general objective they were created for. On the scale 1 indicated "not relevant"; 2 indicated "somewhat relevant"; 3 indicated "quite relevant"; and 4 indicated "highly relevant".

To quantify the content validity of the educational module and to represent evidence of the validation, the approach of content validity index (CVI) was used. Two forms of CVI were calculated, item-level content validity index (I-CVI) and scale-level content validity index (S-CVI) [27]. The I-CVI indicates the proportion of experts giving an item a relevance rating of 3 or 4 [27]. I-CVI for each item is the average of experts who rated the item as relevant (number of experts who rated the item as relevant divided by the total number of experts). While the S-CVI can be based on the I-CVI (S-CVI/Ave) which refers to the average of I-CVI scores across all items on the scale, or based on the universal agreement (S-CVI/UA) which is the average of the universal agreement (UA) scores, where a score of 1 is given to the items that achieved 100% of experts agreeing, otherwise, the universal agreement (UA) score is given as 0 [27]. Proportion relevance was also calculated. It refers to the average of relevant scores across all experts [27]. The acceptable cut-off score of CVI is at least 0.83 when the number of experts is at least six which is the case in this validation [27].

**Face validation by target population:** Face validation of the module was conducted to ensure that the module is accurately suitable for the targeted population for whom it was developed and that they are represented in the process of its evaluation. Sixteen mothers of school-age children participated in evaluating the educational module. The minimum acceptable number of respondents for face validation is ten [29]. The inclusion criteria were parents of children between 8 and 12 years of age who are literate in the Arabic language. The evaluation was conducted in face-to-face meetings with the participants. They were asked to

read the module and indicate the vocabulary they found difficult and to suggest alternative vocabulary that they thought was more understandable. They were also asked to evaluate the adequacy of the illustrations and overall presentation. They had to rate each message of the 26 messages in the module on a scale of 1–4. The response options were: 1. not clear and not understandable, 2. somewhat clear and understandable, 3. quite clear and understandable, and 4. very clear and understandable. They were also given an evaluation questionnaire with a total of 17 multiple-choice questions about the overall structure and content clarity and comprehensibility with free space for any comments or suggestions they might have.

Like the content validity, quantification and representation of evidence from the face validation were done through calculations but using the approach of face validity index (FVI). Two forms of FVI were calculated, item-level face validity index (I-FVI) and scale-level face validity index (S-FVI) [29]. The I-FVI indicates the proportion of raters giving an item a clarity and comprehension rating of 3 or 4 [29]. I-FVI for each item is the average of raters who rated the item as relevant (number of raters who rated the item as relevant divided by the total number of raters). While the S-FVI can be based on the I-FVI (S-FVI/Ave) which refers to the average of I-FVI scores across all items on the scale, or based on the universal agreement (S-FVI/UA) which is the average of the universal agreement (UA) scores where a score of 1 is given to the items that achieved 100% of raters agreeing, otherwise, the universal agreement (UA) score is given as 0 [29]. Proportion clarity and comprehension was also calculated. It refers to the average of clear and comprehensible scores across all raters [29]. The acceptable cut-off score of FVI is at least 0.83 for the number of participants in this validation [29].

**Data analysis:** Data analysis focused on quantifying the content and face validity of the HEM. Calculations were performed manually and verified using Microsoft Excel. Content validity was assessed for item relevance using I-CVI and S-CVI (S-CVI/Ave and S-CVI/UA), while face validity was assessed for clarity and comprehensibility using I-FVI and S-FVI (S-FVI/Ave and S-FVI/UA).

- I-CVI and I-FVI: Ratings of 3 or 4 were considered valid. The respondents for the I-CVI are the experts and for the I-FVI are the parents:

$$I - CVI \ (or \ I - FVI) \ = \ \frac{\text{Number of respondents rating the item as 3 or 4}}{\text{Number of respondents}}$$

- S-CVI/Ave and S-FVI/Ave: Average of all item-level validity scores (I-CVI and I-FVI) across all items:

$$S - CVI/Ave \ (or \ S - FVI/Ave) \ = \ \frac{\text{Sum of all item} - \text{level validity scores}}{\text{Number of items}}$$

- S-CVI/UA and S-FVI/UA: Each item achieved universal agreement (scored 1) if all respondents rated it 3 or 4; otherwise, it was scored 0:

$$S - CVI/UA \ (or \ S - FVI/UA) \ = \ \frac{\text{Number of items with universal agreement}}{\text{Number of items}}$$

- Proportions and average proportions were also calculated for the relevance and clarity and comprehensibility criteria:

$$\text{Proportion} = \frac{\text{Number of respondents rating the item as 3 or 4}}{\text{Number of items}}$$

$$\text{Average proportion} \ = \ \frac{\text{Proportion}}{\text{Number of respondents}}$$

A cut-off of ≥0.83 indicated acceptable content and face validity. Qualitative feedback from experts and parents was used to refine the module.

The following flowchart (Fig 1) summarizes the methodological steps undertaken in the development and validation of the health education module, presented across the two main phases of the study:

## Results

### Module development and modification

The HEM was developed and designed using the main themes that resulted from the literature and guidelines review. These themes were presented through 26 infographic messages, as outlined in Table 1.

All the messages successfully passed through the validation process with only a few comments on language and appearances that were taken into consideration. Only one message about the concept of the "Food Pyramid" was removed after the validation process although the message itself with its information met the cut-off for acceptance, but the comments of the expert evaluators were valid and appealing. The evaluators suggested that it is an old concept and proposed that the message discussing the "MyPlate" food guide [30] is a better and more up-to-date replacement. The final module consisted of a total of 25 messages that were assessed for the use of words, information accuracy, sentence formation, display and illustrations on expert and target population levels.

**Content validation.** The entire module underwent a thorough review and evaluation by the six participating experts. All the experts provided ratings indicating the content's relevance of each message, giving each a score of 1–4 on the

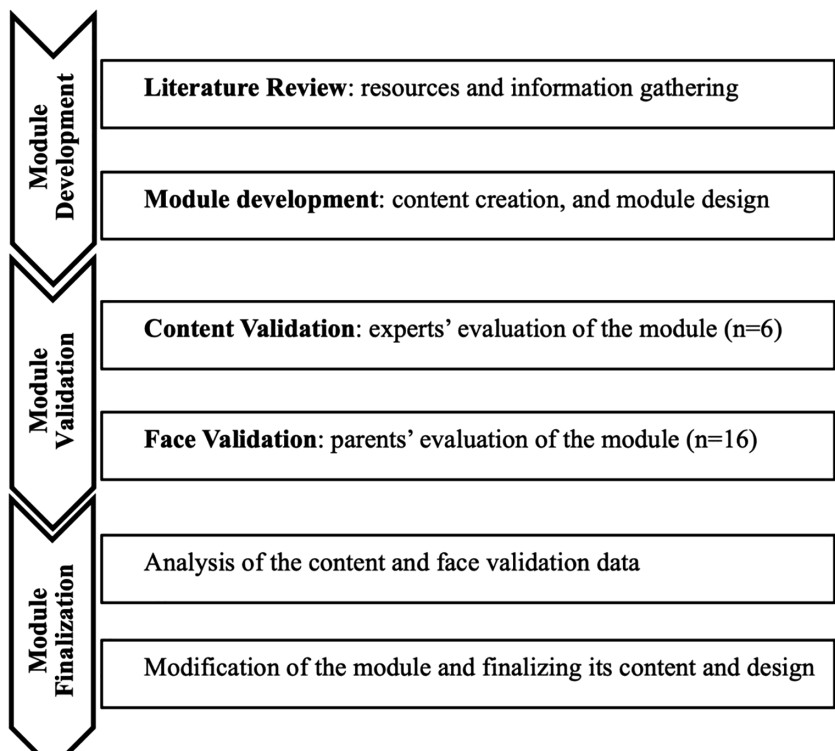

**Fig 1. Flowchart of the study.**

**Table 1. Topics and content description of the educational module messages.**

|  | Message Title | Message content |
|---|---|---|
| 1 | Childhood Obesity | Definition of childhood obesity and complications related to it, and the actions that need to be taken |
| 2 | Weight and height measurement | Steps to correctly and accurately measure a child's weight and height |
| 3 | Changing old habits | Steps to changing bad habits related to food and health with good habits |
| 4 | General guidelines for healthy eating | Introduction to food groups and what to be chosen and avoided from each |
| 5 | Carbohydrates | Importance, serving sizes for children, and healthy and unhealthy sources |
| 6 | Dietary fiber | Definition, importance, recommendation, and sources of fiber |
| 7 | Fruit and vegetables | Importance, serving sizes for children, and healthy and unhealthy sources |
| 8 | Dairy products | Importance, serving sizes for children, and healthy and unhealthy sources |
| 9 | Meat products and alternatives | Importance, serving sizes for children, and healthy and unhealthy sources |
| 10 | Sugar, salt, and saturated fat | Emphases on health risks related to each and the food items rich in them |
| 11 | Water and fluid intake | Recommendations, healthy and unhealthy sources, and suggestions |
| 12 | Food Pyramid | Display of the pyramid and explaining the concept and the recommendations |
| 13 | MyPlate | Display of the plate and explaining how to use it to ensure balanced meals |
| 14 | Serving sizes using hands and house-hold items | Visual explanation of how to use hands and household items to estimate serving sizes |
| 15 | Healthy alternatives | Examples of food items and cooking methods and healthy replacements |
| 16 | Healthy school lunchbox | Steps and ideas to improve school lunchbox and healthy options to pack |
| 17 | How to read nutrition facts and food labels | Display of nutrition facts card with explanation, and steps to smart grocery shopping label reading |
| 18 | Meals of the day | Number of meals a day for a child and importance and what should include |
| 19 | Home environment | Role of parents around the house and how to influence child's habits |
| 20 | Child involvement | Importance of child's role in meal planning and awareness of their eating |
| 21 | Food as reward | The importance of food not being a way of showing love or reward for children |
| 22 | Importance of physical activity | Role and importance of physical activity in child's health improvement |
| 23 | Physical activity recommendations | Recommendations and ideas to increase child's physical activity |
| 24 | Time of physical activity | Ideas to involve the child in planning the time and kind of physical activity |
| 25 | Screen time and sedentary behaviour | Effect of electronics use on a child and the role of parents in limiting it |
| 26 | Why does good sleep matter? | Recommendations and comparison between well-sleeping and sleep-deprived children |

rating scales as shown in Table 2. Ratings of 3 and 4 were merged into one group as both reflected the evaluation of each item as relevant, although minor revisions were suggested for items given a rate of 3.

These revisions aimed to improve specific areas of the module, such as rephrasing confusing statements, including additional explanations for clarity, and incorporating suggested examples to reinforce certain messages. Subsequent modifications to the module and its messages were made to these recommendations.

The open-ended questions at the end of the questionnaire regarding the experts' likes, dislikes, comments, and suggestions led to the removal of the "food pyramid" message as explained earlier. Their overall opinion was positive as four out of the six experts expressed their admiration for the module's comprehensiveness of the nutritional, behavioral, psychological, and social aspects related to childhood obesity and health in general. The complete results of the content validity are summarized in Table 2. Experts also rated the module based on seven aspects related to the module. Table 3 summarizes the results of the experts' evaluation of each aspect.

**Face validation.** The messages of the module were evaluated by 16 mothers of schoolchildren at the age of 8–12 years. The evaluations were conducted in individual meetings with the participants. All the mothers gave ratings that indicate the clarity and comprehensibility of each message. The results of the face validation of all the messages are

**Table 2. Content validity index for the educational module by the expert panel (n = 6).**

| Item | Relevant (Rating 3 or 4) | Not Relevant (Rating 1 or 2) | I-CVI[†] | UA[‡] |
|---|---|---|---|---|
| Message 1 | 6 | 0 | 1 | 1 |
| Message 2 | 6 | 0 | 1 | 1 |
| Message 3 | 6 | 0 | 1 | 1 |
| Message 4 | 5 | 1 | 0.83 | 0 |
| Message 5 | 6 | 0 | 1 | 1 |
| Message 6 | 6 | 0 | 1 | 1 |
| Message 7 | 5 | 1 | 0.83 | 0 |
| Message 8 | 6 | 0 | 1 | 1 |
| Message 9 | 6 | 0 | 1 | 1 |
| Message 10 | 6 | 0 | 1 | 1 |
| Message 11 | 5 | 1 | 0.83 | 0 |
| Message 12 | 6 | 0 | 1 | 1 |
| Message 13 | 6 | 0 | 1 | 1 |
| Message 14 | 5 | 1 | 0.83 | 0 |
| Message 15 | 6 | 0 | 1 | 1 |
| Message 16 | 6 | 0 | 1 | 1 |
| Message 17 | 6 | 0 | 1 | 1 |
| Message 18 | 6 | 0 | 1 | 1 |
| Message 19 | 6 | 0 | 1 | 1 |
| Message 20 | 6 | 0 | 1 | 1 |
| Message 21 | 6 | 0 | 1 | 1 |
| Message 22 | 6 | 0 | 1 | 1 |
| Message 23 | 6 | 0 | 1 | 1 |
| Message 24 | 6 | 0 | 1 | 1 |
| Message 25 | 6 | 0 | 1 | 1 |
| Message 26 | 6 | 0 | 1 | 1 |
| | | S-CVI/Ave[§] | 0.97 | |
| | | S-CVI/UA[§] | | 0.85 |
| **Proportion Relevance** | | | 5.84 | |
| **The average proportion of items judged as relevant across the six experts** | | | 0.97 | |

[†] I-CVI = Item-level Content Validity Index.

[‡] UA = Universal Agreement.

[§] S-CVI = Scale-level Content Validity Index.

shown in Table 4. Ratings of 3 and 4 were merged into one group as both reflected the evaluation of each item as clear and comprehensible, although very few comments were given for messages rated as 3. These comments reflected the raters' opinions regarding the amount of explanation for information given in each of these messages. They suggested giving more explanation and backing it up with more examples to make some of the ideas clearer. These suggestions were taken into consideration while modifying the module.

The participants also answered the 17 multiple-choice questions that were asked in a questionnaire. The results are presented in Table 5. At the end of the questionnaire, participants were given a few open-ended questions regarding their likes, dislikes, comments, and suggestions they would like to add. Most of them were satisfied with expressing their opinions through the responses they gave to the multiple-choice questions. Only two of them gave some extra suggestions regarding giving examples of complete diet plans. One of them even suggested supplementary videos to visually

**Table 3. Content and face validity aspects evaluated by the expert panel (n = 6).**

| Criteria | Item Description | Relevant (Rating 3 or 4) | Not Relevant (Rating 1 or 2) | I-CVI† | UA‡ |
|---|---|---|---|---|---|
| **Scientific Accuracy** | contents are in agreement with the current knowledge | 6 | 0 | 1 | 1 |
| | recommendations are necessary and are correctly approached | 6 | 0 | 1 | 1 |
| | S-CVI/Ave§ | | | 1 | |
| **Content** | recommendation about the desired behavior is satisfactory | 6 | 0 | 1 | 1 |
| | there is no unnecessary information | 6 | 0 | 1 | 1 |
| | important points are reviewed | 6 | 0 | 1 | 1 |
| | S-CVI/Ave§ | | | 1 | |
| **Literary Presentation** | language is neutral (no comparative adjectives, promotion or false appeals) | 6 | 0 | 1 | 1 |
| | language is explanatory | 6 | 0 | 1 | 1 |
| | material promotes and encourages treatment adherence following evaluation of the benefits and risks | 6 | 0 | 1 | 1 |
| | majority of the vocabulary is composed of common words | 6 | 0 | 1 | 1 |
| | vocabulary is composed of simple words | 6 | 0 | 1 | 1 |
| | language is adequate for the general population | 6 | 0 | 1 | 1 |
| | ideas are concisely expressed | 6 | 0 | 1 | 1 |
| | planning and sequence of information is consistent, making it easier to predict its flow | 6 | 0 | 1 | 1 |
| | material is reader-friendly (easy to read) | 6 | 0 | 1 | 1 |
| | S-CVI/Ave§ | | | 1 | |
| **Illustrations** | illustrations are simple, appropriate and present an easily understandable outline | 6 | 0 | 1 | 1 |
| | illustrations are familiar to the readers | 6 | 0 | 1 | 1 |
| | illustrations are related to the text (express the desired purpose) | 6 | 0 | 1 | 1 |
| | illustrations are integrated with the text (easily located) | 6 | 0 | 1 | 1 |
| | S-CVI/Ave§ | | | 1 | |
| **Material is sufficiently specific and understandable** | health messages delivered are understandable | 6 | 0 | 1 | 1 |
| | the text provides maximum benefit with the minimization of complications | 6 | 0 | 1 | 1 |
| | it clearly explained how to optimise dietary intake | 6 | 0 | 1 | 1 |
| | headings and subheadings are clear and informative | 6 | 0 | 1 | 1 |
| | use of words or expressions with double meanings does not occur in the text | 6 | 0 | 1 | 1 |
| | content is focused on children (the child is the focus of attention) | 6 | 0 | 1 | 1 |
| | S-CVI/Ave§ | | | 1 | |
| **Legibility and Display Characteristics** | size of the font is adequate | 6 | 0 | 1 | 1 |
| | style of the font is adequate | 6 | 0 | 1 | 1 |
| | spacing between words is adequate | 6 | 0 | 1 | 1 |
| | length of the lines is adequate | 6 | 0 | 1 | 1 |
| | spacing between lines is adequate | 6 | 0 | 1 | 1 |
| | use of bold characters and bullet points draw attention to specific points or key content | 6 | 0 | 1 | 1 |
| | adequate use of blank space reduces the overcrowded appearance | 6 | 0 | 1 | 1 |
| | good combination of colours used | 6 | 0 | 1 | 1 |
| | use of colors draws the attention of readers | 6 | 0 | 1 | 1 |
| | spacing between paragraphs is adequate | 6 | 0 | 1 | 1 |
| | format of the material is adequate | 6 | 0 | 1 | 1 |

*(Continued)*

**Table 3.** (Continued)

| Criteria | Item Description | Relevant (Rating 3 or 4) | Not Relevant (Rating 1 or 2) | I-CVI[†] | UA[‡] |
|---|---|---|---|---|---|
| | | S-CVI/Ave[§] | | 1 | |
| Quality of Information | information is integrated into the local culture | 5 | 1 | 0.83 | 0 |
| | information is updated | 5 | 1 | 0.83 | 0 |
| | it is adapted to the current culture | 6 | 0 | 1 | 1 |
| | material enables the reader to undertake the desired actions | 6 | 0 | 1 | 1 |
| | material helps the reader to prevent potential problems | 6 | 0 | 1 | 1 |
| | material allows the reader to achieve the maximum benefits possible | 6 | 0 | 1 | 1 |
| | | S-CVI/Ave[§] | | 0.94 | |
| | | Overall S-CVI/Ave[§] | | 0.99 | |
| | | Overall S-CVI/UA[§] | | | 0.95 |
| Proportion Relevance | | 5.95 | | | |
| The average proportion of items judged as relevant across the six experts | | 0.99 | | | |

[†] I-CVI = Item-level Content Validity Index.

[‡] UA = Universal Agreement.

[§] S-CVI = Scale-level Content Validity Index.

talk about diet plans. These suggestions although didn't affect the validity of the module were valuable as they inspired a further step to be taken in the future by adding supplementary visual aids and diet plans to the educational module to make it a complete program.

After all the evaluations were processed and all the items in the module met the acceptance cut-off to become valid and as all suggestions were taken into consideration and minor adjustments were made accordingly, the developed educational module can be considered a valid tool that can be used in interventions to educate parents and improve dietary intake and consequently the health of schoolchildren.

## Discussion

The objective of this study was to develop and validate a HEM targeted specifically to parents of schoolchildren affected by overweight or obesity. The uniqueness of this module is in its ability to work as a specific tool for a specific targeted population to manage childhood obesity, while at the same time can be generalized to the whole population as an educational material that can help and guide parents in the improvement of the health of all children and still be very effective. That was made possible by making this module focus on educating parents about healthy lifestyle and recommended dietary guidelines for children rather than weight loss. It focuses on changing habits and expanding the parents' understanding of their children's needs.

Prior studies have repeatedly emphasized the significant influence of parents on children's dietary habits and lifestyle behaviors [10–12,17], yet few interventions in the UAE have produced structured, parent-centered educational materials. Unlike earlier work, which often identified gaps in parental knowledge without providing a validated tool to address them [3,17], the present module fills this void by offering a comprehensive, evidence-based resource.

This module was intentionally developed to emphasize healthy lifestyle promotion rather than strict weight loss, a direction supported by research showing that parent-led lifestyle interventions yield more sustainable outcomes in children than weight-focused programs [9,11]. Consistent with findings from Lambrinou et al., integrating nutrition guidance, behavior modification strategies, and family-based approaches increases intervention effectiveness [9]. Similarly, previous

**Table 4. Face validity index for the educational module by the general audience (n = 16).**

| Item | Clear & Comprehensible (Rating 3 or 4) | Not Clear & Comprehensible (Rating 1 or 2) | I-FVI[†] | UA[‡] |
|---|---|---|---|---|
| Message 1 | 16 | 0 | 1 | 1 |
| Message 2 | 16 | 0 | 1 | 1 |
| Message 3 | 16 | 0 | 1 | 1 |
| Message 4 | 16 | 0 | 1 | 1 |
| Message 5 | 16 | 0 | 1 | 1 |
| Message 6 | 16 | 0 | 1 | 1 |
| Message 7 | 16 | 0 | 1 | 1 |
| Message 8 | 16 | 0 | 1 | 1 |
| Message 9 | 16 | 0 | 1 | 1 |
| Message 10 | 16 | 0 | 1 | 1 |
| Message 11 | 16 | 0 | 1 | 1 |
| Message 12 | 16 | 0 | 1 | 1 |
| Message 13 | 16 | 0 | 1 | 1 |
| Message 14 | 14 | 2 | 0.88 | 0 |
| Message 15 | 16 | 0 | 1 | 1 |
| Message 16 | 16 | 0 | 1 | 1 |
| Message 17 | 15 | 1 | 0.94 | 0 |
| Message 18 | 16 | 0 | 1 | 1 |
| Message 19 | 15 | 1 | 0.94 | 0 |
| Message 20 | 15 | 1 | 0.94 | 0 |
| Message 21 | 16 | 0 | 1 | 1 |
| Message 22 | 16 | 0 | 1 | 1 |
| Message 23 | 16 | 0 | 1 | 1 |
| Message 24 | 16 | 0 | 1 | 1 |
| Message 25 | 16 | 0 | 1 | 1 |
| Message 26 | 16 | 0 | 1 | 1 |
| | | **S-FVI/Ave[§]** | 0.99 | |
| | | **S-FVI/UA[§]** | | 0.85 |
| **Proportion Relevance** | | | 15.8 | |
| **The average proportion of items judged as clear and Comprehensible across the 16 raters** | | | 0.99 | |

† I-FVI = Item-level Face Validity Index.

‡ UA = Universal Agreement.

§ S-FVI = Scale-level Face Validity Index.

systematic reviews have highlighted the need for interventions that strengthen parental self-efficacy and enable long-term habit formation [11,12], which aligns with the guiding principles incorporated into this module.

Another key consideration in the module's development was the mode of communication. While printed educational material can still be a very effective tool to spread knowledge, technology and social media have changed the concepts of communication and information accessibility. People can access so much information related to health in a very convenient way regardless of the source or the scientific evidence of the information. Earlier studies have reported that interventions delivered through flexible, technology-based platforms can enhance parental engagement and accessibility [13]. At the same time, parents in the UAE have expressed a preference for practical, visually appealing, and easily shareable educational formats [17].

**Table 5. Face validity aspects evaluated by the general audience (n = 16).**

| Questions | Given Choices | Number of Agreements |
|---|---|---|
| **1. What do you like about this health education module?** | | |
| | Pictures | 12 |
| | Content | 15 |
| | The use of WhatsApp as an educational method | 7 |
| | Others | 0 |
| **2. Do you understand the health message conveyed?** | | |
| | Strongly understand | 16 |
| | Understand | 0 |
| | Not really | 0 |
| **3. What makes you not understand the health messages given?** | | |
| | Words that are difficult to understand | 0 |
| | Sentences that are difficult to understand | 0 |
| | Inappropriate pictures | 0 |
| | Others | 0 |
| **4. What factors influence your understanding of the message conveyed?** | | |
| | Words | 8 |
| | Sentences | 12 |
| | Pictures | 9 |
| | Others | 3 |
| **5. What is needed to increase your understanding?** | | |
| | Add more pictures | 3 |
| | Add more tables and diagrams | 3 |
| | Use simple sentences | 1 |
| | Reduce the use of scientific sentences | 3 |
| | Nothing | 8 |
| **6. How do you rate the use of graphics/pictures in the messages?** | | |
| | Important | 16 |
| | Not important | 0 |
| | Not sure | 0 |
| **7. How is the layout of the pictures and diagrams used?** | | |
| | Appropriate | 16 |
| | Inappropriate | 0 |
| | Not sure | 0 |
| **8. How do you rate the messages' writing?** | | |
| | Very good | 16 |
| | Good | 0 |
| | Bad | 0 |
| | Very bad | 0 |
| **9. How do you rate the colours used in the message?** | | |
| | Very good | 16 |
| | Good | 0 |
| | Bad | 0 |
| | Very bad | 0 |
| **10. Does the colour combination used interest you?** | | |
| | Yes | 16 |
| | No | 0 |
| | Not sure | 0 |
| **11. Is the information in this educational material adequate?** | | |
| | Yes | 15 |
| | No | 0 |
| | Not sure | 1 |

*(Continued)*

**Table 5.** (Continued)

| Questions | Given Choices | Number of Agreements |
|---|---|---|
| **12. Is there anything new you've learned from this educational material?** | | |
| | Yes | 13 |
| | No | 3 |
| | Not sure | 0 |
| **13. What is your level of acceptance of the messages?** | | |
| | Strongly accept | 8 |
| | Accept | 8 |
| | Not accept | 0 |
| **14. Is the content in this educational material appropriate to share with parents?** | | |
| | Yes | 15 |
| | No | 0 |
| | Not sure | 1 |
| **15. Do you think this educational material can help prevent obesity in children?** | | |
| | Yes | 16 |
| | No | 0 |
| **16. Is there anything that needs to be added to this educational material?** | | |
| | Yes | 2 |
| | No | 14 |
| **17. Is there anything that needs to be removed from this educational material?** | | |
| | Yes | 0 |
| | No | 16 |

This module was designed to be a flexible tool that can be printed into a booklet or delivered online as an e-book or through social media as individual messages. This flexibility will help in distributing this scientifically evaluated evidence-based educational material to reach more people and addressing these modern communication trends more effectively than previous interventions that relied primarily on static, text-heavy materials [3]. It can be employed in public health strategies or used as part of schools' health education programs directed to students' parents, or any other settings that aim to health educate parents and to combat childhood obesity whether face-to-face or virtually.

For research however, the module was evaluated and validated by experts and the target population to create a good quality comprehensive educational tool to be used in current and future research studies. Previous UAE-based studies have largely focused on assessing prevalence, parental perceptions, or barriers to healthy behaviors [3,17], but few have developed validated educational materials. This study contributes to filling that gap by incorporating both expert validation and community-level face validation, strengthening the module's credibility as a research tool.

This study assessed the content and face validity of the module ensuring its suitability to meet the objectives behind developing it. The expert participants in the validation of this module rated the relevance of its content above 97%. This aligns with recommendations from validation research, which emphasizes the importance of multi-domain assessment, including accuracy, clarity, cultural appropriateness, and behavioral relevance, to strengthen educational materials [27,28]. Moreover, the experts described the module as complete and consistent as it brings together different aspects related to children's health such as dietary guidelines, physical activity, home behavior change, and family engagement, which reflects the multidimensional approach recommended by previous intervention frameworks such as those described in Feel4Diabetes and other school-based family-involved programs [9].

The participants from the target population also had a valuable role in the validation of this module. They rated its clarity and comprehensibility above 99%, indicating strong alignment with findings from Zuarub et al., who reported that

parents in the UAE prefer simple, visually guided, culturally relevant educational content [17]. Many existing interventions fail to meet these preferences, often being too technical or insufficiently adapted to family dynamics [15,16].

Compared to previous parental-education interventions in the region, the present module demonstrates stronger methodological rigor through the use of formal content and face validation, an approach that has been limited or absent in earlier studies [3,11,17]. Previous research in the UAE has consistently documented parental knowledge gaps, challenges in supporting healthy behaviors, and the need for structured, practical guidance [2–4,17], yet few interventions provide standardized, evidence-based tools specifically designed for parents and validated for clarity, cultural relevance, and usability [11,15,17]. By integrating behavioral aspects, particularly principles aligned with parental influence on child behavior and family-based approaches [10–12], addressing cultural appropriateness through tailored examples and messaging [15,16], and covering a comprehensive set of topics spanning nutrition, physical activity, home environment, and behavior change, the developed module addresses key shortcomings noted in earlier work [3,11,17].

With the module and its items exceeding the cut-off of acceptance for content and face validation, and after the consideration of the evaluators' suggestions and the minor adjustments that were done accordingly, the developed educational module can be considered a valid tool that can be used in research interventions or health education endeavors to reduce childhood obesity and to improve children's health in general. This aligns with global and regional recommendations emphasizing the importance of culturally sensitive, family-centered approaches in combating childhood obesity [8,9].

However, this study only assessed the content and face validity of the module. Other forms of validation would have provided a wider range of evidence to strengthen the evaluation of the module [11,14]. Therefore, further research is recommended to include more validation tests and to test the reliability and effectiveness of this module in improving knowledge related to children's health among parents.

## Conclusion

This study produced a culturally adapted, scientifically grounded health education module that underwent systematic content and face validation. With high expert and parent agreement on clarity, relevance, and usability, the module is well suited for health-education initiatives targeting parents of schoolchildren with overweight and obesity in the UAE. It can be incorporated into school and community-based programs and used in future research. Further studies are needed to evaluate its reliability, long-term impact, and effectiveness in improving parental practices and children's health outcomes.

## Acknowledgments

The authors would like to thank all the experts who participated in the validation process. The authors would also like to thank all the participants for their time and contribution to this study. The efforts of the researchers and all those involved in this project are greatly appreciated.

## Author contributions

**Conceptualization:** Heba S. M. Mustafaalsaafin, Hamid Jan B. Jan Mohamed, Hafzan Yusoff, Hayder Hasan.

**Data curation:** Heba S. M. Mustafaalsaafin, Mona Hashim, Hadia Radwan, Leila Cheikh Ismail, Maysm N. Mohamad, Heba F. Almassri.

**Formal analysis:** Heba S. M. Mustafaalsaafin, Hamid Jan B. Jan Mohamed, Hafzan Yusoff, Mona Hashim, Hadia Radwan, Hayder Hasan.

**Funding acquisition:** Hayder Hasan.

**Investigation:** Heba S. M. Mustafaalsaafin, Leila Cheikh Ismail, Maysm N. Mohamad, Heba F. Almassri.

**Methodology:** Heba S. M. Mustafaalsaafin, Hafzan Yusoff.

**Project administration:** Heba S. M. Mustafaalsaafin.

**Resources:** Heba S. M. Mustafaalsaafin, Hamid Jan B. Jan Mohamed.

**Supervision:** Hamid Jan B. Jan Mohamed, Leila Cheikh Ismail, Hayder Hasan.

**Validation:** Heba S. M. Mustafaalsaafin.

**Writing – original draft:** Heba S. M. Mustafaalsaafin, Hamid Jan B. Jan Mohamed, Hayder Hasan.

**Writing – review & editing:** Heba S. M. Mustafaalsaafin, Hamid Jan B. Jan Mohamed, Hafzan Yusoff, Mona Hashim, Hadia Radwan, Leila Cheikh Ismail, Maysm N. Mohamad, Heba F. Almassri, Hayder Hasan.

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
