## [Decision Letter · Decision Letter 0]

15 Oct 2025

PONE-D-25-37277Development and validation of a health education module for parents of schoolchildren with overweight and obesity in the UAEPLOS ONE

Dear Dr. Hasan

Thank you for submitting your manuscript to PLOS ONE. After careful consideration, we feel that it has merit but does not fully meet PLOS ONE’s publication criteria as it currently stands. Therefore, we invite you to submit a revised version of the manuscript that addresses the points raised during the review process.

We look forward to receiving your revised manuscript.

Kind regards,

Aamna AlShehhi, PhD

Academic Editor

PLOS ONE

**Journal Requirements:**

1. When submitting your revision, we need you to address these additional requirements. Please ensure that your manuscript meets PLOS ONE's style requirements, including those for file naming. The PLOS ONE style templates can be found at https://journals.plos.org/plosone/s/file?id=wjVg/PLOSOne_formatting_sample_main_body.pdf and https://journals.plos.org/plosone/s/file?id=ba62/PLOSOne_formatting_sample_title_authors_affiliations.pdf 2. If the reviewer comments include a recommendation to cite specific previously published works, please review and evaluate these publications to determine whether they are relevant and should be cited. There is no requirement to cite these works unless the editor has indicated otherwise.

Reviewers' comments:

Reviewer's Responses to Questions

**Comments to the Author**

1. Is the manuscript technically sound, and do the data support the conclusions?

Reviewer #1: Yes

Reviewer #2: Yes

Reviewer #3: Yes

2. Has the statistical analysis been performed appropriately and rigorously?

Reviewer #1: Yes

Reviewer #2: Yes

Reviewer #3: Yes

3. Have the authors made all data underlying the findings in their manuscript fully available?

Reviewer #1: Yes

Reviewer #2: Yes

Reviewer #3: Yes

4. Is the manuscript presented in an intelligible fashion and written in standard English?

Reviewer #1: Yes

Reviewer #2: Yes

Reviewer #3: Yes

5. Review Comments to the Author

**Reviewer #1:** Please use the space provided to explain your answers to the questions above. You may also include additional comments for the author, including concerns about dual publication, research ethics, or publication ethics. (Please upload your review as an attachment if it exceeds 20,000 characters) (Limit 200 to 20000 Characters)

I don't have additional comments , thanks.

**Reviewer #2:** The manuscript addresses an important public health concern and presents a relevant and well-structured study. Overall, the paper is technically sound, the methodology is appropriate, and the findings are clearly presented. Below are my detailed comments:

Technical Soundness and Rigor:- The study design, consisting of two phases (module development and validation), is appropriate and clearly described.

Statistical analysis was performed according to recommended standards, and results are presented transparently.

Data Availability: The authors have complied with PLOS ONE’s data availability policy. All relevant data are included within the manuscript tables and the supporting information files. This ensures reproducibility and transparency.

Presentation and Language: The manuscript is presented in a clear and intelligible manner, with standard scientific English throughout.

Minor issues to correct at revision: On page 13, “health massages” should be corrected to “health messages.”

A few sentences could be smoothed for readability (e.g., “step started with a background reading” → “the step began with a background literature review”).

The abstract and conclusion include some repetition; these could be slightly condensed without loss of clarity.

Strengths of the Manuscript: The study is culturally relevant and fills a gap in the literature by focusing on parental involvement in the UAE context.

The educational module is flexible and adaptable for different platforms (booklet, online, social media), which increases its applicability and potential impact.

The authors have been transparent in acknowledging the limitation that only content and face validity were tested, and they appropriately recommend further reliability and effectiveness studies.

Ethics and Compliance: Ethical approval and informed consent are clearly reported.

**Reviewer #3:** The article was reviewed under the title "Development and validation of a health education module for parents of schoolchildren with overweight and obesity in the UAE".

Overall, the focus of the study is good. However, I would like to offer several recommendations that authors may find useful in the process of revising their manuscript:

1- Given the urgency of the issue, a review of previous research is needed.

2- Considering that it is stated at the end of the introduction that this article was written with the aim of filling an important gap in the literature, in addition to describing previous research, it is necessary to describe and introduce the existing gaps.

3- For better readability, it is necessary to express the overall workflow in the form of an algorithm, flowchart, or figure.

4- Given that one of the stages of the work was the development of the health education module, it is necessary to explain the basic model.

5- Given the statistical nature of the work, a more comprehensive statistical analysis of the results is needed.

6- The number of the statistical population is not specified correctly. Only the time frame for data collection is mentioned.

7- The article states that this method will serve as a comprehensive, high-quality educational tool for use in current and future research studies. In this regard, no comparison between the results obtained with previous research has been made to substantiate this claim.

6. PLOS authors have the option to publish the peer review history of their article (what does this mean?). If published, this will include your full peer review and any attached files.

Reviewer #1: **Yes:**Suzan Abdel-Rahman

Reviewer #2: **Yes:**Tilahun Shiferaw Shibeshi

Reviewer #3: No

---

## [Author Response · Author response to Decision Letter 1]

27 Nov 2025

Dear academic editor and respected reviewers,

Thank you for taking the time to review this paper and for your thorough feedback. The comments you provided have been addressed below.

Journal Requirements

The style and format of the paper have been reviewed to make sure it meets the journal requirements. Minor format editing have been done to the title of the paper and the headings.

Reviewers Comments to the Authors

Reviewer #2:

“The manuscript addresses an important public health concern and presents a relevant and well-structured study. Overall, the paper is technically sound, the methodology is appropriate, and the findings are clearly presented. Below are my detailed comments:

Technical Soundness and Rigor: The study design, consisting of two phases (module development and validation), is appropriate and clearly described.

Statistical analysis was performed according to recommended standards, and results are presented transparently.

Data Availability: The authors have complied with PLOS ONE’s data availability policy. All relevant data are included within the manuscript tables and the supporting information files. This ensures reproducibility and transparency.

Presentation and Language: The manuscript is presented in a clear and intelligible manner, with standard scientific English throughout.

Strengths of the Manuscript: The study is culturally relevant and fills a gap in the literature by focusing on parental involvement in the UAE context.

The educational module is flexible and adaptable for different platforms (booklet, online, social media), which increases its applicability and potential impact.

The authors have been transparent in acknowledging the limitation that only content and face validity were tested, and they appropriately recommend further reliability and effectiveness studies.

Ethics and Compliance: Ethical approval and informed consent are clearly reported.”

Response:

Thank you for your kind and detailed feedback. The minor issues mentioned in your comment have been addressed as follows:

1. On page 13, “health massages” should be corrected to “health messages.”

- The spilling mistake in “health messages” has been corrected (page 15).

2. A few sentences could be smoothed for readability (e.g., “step started with a background reading” → “the step began with a background literature review”).

- The sentence has been modified as suggested (page 6 / line 113-114).

3.  The abstract and conclusion include some repetition; these could be slightly condensed without loss of clarity.

- The abstract and the conclusion have been rephrased and condensed to make them clearer and to avoid repetition (page 2, 24).

Reviewer #3:

“The article was reviewed under the title "Development and validation of a health education module for parents of schoolchildren with overweight and obesity in the UAE".

Overall, the focus of the study is good. However, I would like to offer several recommendations that authors may find useful in the process of revising their manuscript.”

Response:

Thank you for your feedback and the useful recommendations. Your comments have been addressed as follows:

1. Given the urgency of the issue, a review of previous research is needed.

- A paragraph was inserted in the introduction section reviewing previous studies (page 4-5 / line 78-90).

2. Considering that it is stated at the end of the introduction that this article was written with the aim of filling an important gap in the literature, in addition to describing previous research, it is necessary to describe and introduce the existing gaps.

- The gaps and the need for this study were also described in the new paragraph inserted in the introduction section to review previous studies (page 4-5 / line 78-90).

3. For better readability, it is necessary to express the overall workflow in the form of an algorithm, flowchart, or figure.

- A flowchart has been added to the end of the methods section with a summary of the main phases of this study (page 10 / line 223-226). The actual figure is uploaded as a separate file.

4. Given that one of the stages of the work was the development of the health education module, it is necessary to explain the basic model.

- A paragraph was added to the health education module development phase section under methods emphasizing the use of the health belief model in the designing of the module (page 6 / line 128-134).

5. Given the statistical nature of the work, a more comprehensive statistical analysis of the results is needed.

- The description of the statistical analysis was indicated within the explanation of the used approaches of CVI and FVI. However, to make it clearer and more statistically formatted, a subheading (Data analysis) was added towards the end of the methods section, that explains the details of the conduction of data analysis and the calculations (page 9-10 / line 202-222).

6. The number of the statistical population is not specified correctly. Only the time frame for data collection is mentioned.

- The number was specified within the methods of each validation. To correctly specify it, it was added under the study design and ethics at the beginning of the methods section (page 6 / line 110-111).

7. The article states that this method will serve as a comprehensive, high-quality educational tool for use in current and future research studies. In this regard, no comparison between the results obtained with previous research has been made to substantiate this claim.

- The discussion section has been modified and integrated with previous research (page 20-23). A paragraph has been added as well with a comparison that substantiate the mentioned claim (page 23 / line 362-373).

---

## [Decision Letter · Decision Letter 1]

2 Jan 2026

PONE-D-25-37277R1Development and validation of a health education module for parents of schoolchildren with overweight and obesity in the UAEPLOS One

Dear Dr. Hasan,

Thank you for submitting your manuscript to PLOS ONE. After careful consideration, we feel that it has merit but does not fully meet PLOS ONE’s publication criteria as it currently stands. Therefore, we invite you to submit a revised version of the manuscript that addresses the points raised during the review process.

We look forward to receiving your revised manuscript.

Kind regards,

Aamna AlShehhi, PhD

Academic Editor

PLOS One

Journal Requirements:

Reviewers' comments:

Reviewer's Responses to Questions

**Comments to the Author**

1. If the authors have adequately addressed your comments raised in a previous round of review and you feel that this manuscript is now acceptable for publication, you may indicate that here to bypass the “Comments to the Author” section, enter your conflict of interest statement in the “Confidential to Editor” section, and submit your "Accept" recommendation.

Reviewer #3: (No Response)

2. Is the manuscript technically sound, and do the data support the conclusions?

Reviewer #3: (No Response)

3. Has the statistical analysis been performed appropriately and rigorously?

Reviewer #3: (No Response)

4. Have the authors made all data underlying the findings in their manuscript fully available?

Reviewer #3: (No Response)

5. Is the manuscript presented in an intelligible fashion and written in standard English?

Reviewer #3: (No Response)

6. Review Comments to the Author

Reviewer #3: (No Response)

7. PLOS authors have the option to publish the peer review history of their article (what does this mean?). If published, this will include your full peer review and any attached files.

Reviewer #3: No

---

## [Author Response · Author response to Decision Letter 2]

9 Jan 2026

Dear academic editor and respected reviewers,

Thank you for taking the time to review the last revisions. Upon reviewing your letter, only one comment was found in the document attached to the letter, and it has been addressed below.

Comment:

1. The article titled " Development and validation of a health education module for parents of schoolchildren with overweight and obesity in the UAE" was reviewed again.

All cases have been corrected by the respected authors, but in one case, " For better readability, it is necessary to express the overall workflow in the form of an algorithm, flowchart, or figure. Despite the authors' response, unfortunately the flowchart in question is not visible in the text of the article.

Response:

Thank you for your comment. The flowchart was attached as a separate file that can be found in the file inventory of the submission as per the journal’s instructions regarding figures. However, in this case and for the respected reviewer to be able to see the flowchart, it has been added in the manuscript within the text (in addition to the separate file) (page 11).

---

## [Decision Letter · Decision Letter 2]

2 Mar 2026

Development and validation of a health education module for parents of schoolchildren with overweight and obesity in the UAE

PONE-D-25-37277R2

Dear Dr. Hayder Hasan,

We’re pleased to inform you that your manuscript has been judged scientifically suitable for publication and will be formally accepted for publication once it meets all outstanding technical requirements.

Kind regards,

Aamna AlShehhi, PhD

Academic Editor

PLOS One

Reviewers' comments:

Reviewer's Responses to Questions

**Comments to the Author**

1. If the authors have adequately addressed your comments raised in a previous round of review and you feel that this manuscript is now acceptable for publication, you may indicate that here to bypass the “Comments to the Author” section, enter your conflict of interest statement in the “Confidential to Editor” section, and submit your "Accept" recommendation.

Reviewer #3: (No Response)

2. Is the manuscript technically sound, and do the data support the conclusions?

Reviewer #3: (No Response)

3. Has the statistical analysis been performed appropriately and rigorously?

Reviewer #3: (No Response)

4. Have the authors made all data underlying the findings in their manuscript fully available?

Reviewer #3: (No Response)

5. Is the manuscript presented in an intelligible fashion and written in standard English?

Reviewer #3: (No Response)

6. Review Comments to the Author

Reviewer #3: (No Response)

7. PLOS authors have the option to publish the peer review history of their article (what does this mean?). If published, this will include your full peer review and any attached files.

Reviewer #3: No

---

## [Editor Report · Acceptance letter]

PONE-D-25-37277R2

PLOS One

Dear Dr. Hasan,

I'm pleased to inform you that your manuscript has been deemed suitable for publication in PLOS One. Congratulations! Your manuscript is now being handed over to our production team.

Kind regards,

on behalf of

Dr Aamna AlShehhi

Academic Editor

PLOS One